# Plasma Levels of a Cleaved Form of Galectin-9 Are the Most Sensitive Biomarkers of Acquired Immune Deficiency Syndrome and Tuberculosis Coinfection

**DOI:** 10.3390/biom10111495

**Published:** 2020-10-30

**Authors:** Shirley T. Padilla, Toshiro Niki, Daisuke Furushima, Gaowa Bai, Haorile Chagan-Yasutan, Elizabeth Freda Telan, Rosario Jessica Tactacan-Abrenica, Yosuke Maeda, Rontgene Solante, Toshio Hattori

**Affiliations:** 1Adult Infectious Disease and Tropical Medicine, San Lazaro Hospital, Manila 1003, Philippines; yelmd10@gmail.com (S.T.P.); rontgenesolante@gmail.com (R.S.); 2Department of Immunology, Kagawa University, Kita-gun, Kagawa 7610793, Japan; 3Department of Drug Evaluation and Informatics, Graduate School of Pharmaceutical Sciences, University of Shizuoka, Shizuoka 422-8526, Japan; dfuru@u-shizuoka-ken.ac.jp; 4Department of Health Science and Social Welfare, Kibi International University, Takahashi 716-8508, Japan; gaowabai@kiui.ac.jp (G.B.); haorile@gjmyemail.gjmyy.cn (H.C.-Y.); 5Mongolian Psychosomatic Medicine Department, International Mongolian Medicine Hospital of Inner Mongolia, Hohhot 010065, China; 6STD AIDS Cooperative Central Laboratory, San Lazaro Hospital, Manila 1003, Philippines; betelan@yahoo.com; 7HIV Department, San Lazaro Hospital, Manila 1003, Philippines; rtactacanabrenica@yahoo.com; 8Viral Section, Department of Microbiology, Faculty of Life Sciences, Kumamoto University, Kumamoto 860-0811, Japan

**Keywords:** AIDS, tuberculosis, galectin-9, osteopontin, truncated Gal-9, severity

## Abstract

Acquired immunodeficiency syndrome (AIDS) complicated with tuberculosis (TB) is a global public issue. Due to the paucity of bacteria in AIDS/TB, blood-based biomarkers that reflect disease severity are desired. Plasma levels of matricellular proteins, such as osteopontin (OPN) and galectin-9 (Gal-9), are known to be elevated in AIDS and TB. Therefore, full-length (FL)-Gal9 and FL-OPN, and their truncated forms (Tr-Gal9, Ud-OPN), and 38 cytokines/chemokines were measured in the plasma of 24 AIDS (other than TB), 49 TB, and 33 AIDS/TB patients. Receiver-operating characteristic analysis was used to screen molecules that could distinguish either between disease and normal group, among each disease group, or between deceased patients and survivors. Selected molecules were further analyzed for significant differences. Tr-Gal9 had the highest ability to differentiate TB from AIDS or AIDS/TB, while Ud-OPN distinguished multidrug resistance (MDR)-TB from non-MDR TB, and extra-pulmonary TB from pulmonary TB. Molecules significantly elevated in deceased patients included; FL-Gal9, Tr-Gal9, interleukin (IL)-1 receptor antagonist, IL-17A and transforming growth factor-α in AIDS; IL-6, granulocyte colony-stimulating factor and monocyte chemotactic protein-1 in TB; and macrophage inflammatory protein-1β in AIDS/TB. From the sensitivity, specificity, and significant elevation, Tr-Gal9 is the best biomarker of inflammation and severity in AIDS and AIDS/TB.

## 1. Introduction

Tuberculosis (TB) is one of the top 10 death causes and the leading death cause from a single infectious agent. Globally, around 10 million people had TB in 2018. There were approximately 251,000 TB deaths among human immunodeficiency virus (HIV)-positive people in 2018 [1]. Since the 1990s, the HIV epidemic has been one of the main causes of TB incidence worldwide. The median TB incidence rate in HIV patients, compared with uninfected people living in the same country, was 22-fold higher in 2016 [2]. In South Africa, the number of deaths caused by acquired immune deficiency syndrome (AIDS) associated with TB in 1990 accounted for 47% of all deaths, and an increase in HIV infections was also noted during this year. Therefore, the invasion of HIV into the TB-endemic area accounts for much of the rise of AIDS/TB [3]. TB incidence is quite high (554) in the Philippines; furthermore, the rate of HIV infection has been alarmingly rising in the subpopulation of men who have sex with men (MSM) in the Philippines. In 2014 alone, there were 6011 newly diagnosed cases of HIV, representing 27% of the total number of cases identified in the Philippines [4]. According to UNAIDS, there was a 203% increase in the rate of new infections between 2010 and 2018, concentrated among MSM, drug users, and sex workers in the Philippines [5]. 

CD4 count reductions within coinfected persons are associated with a poorer granuloma formation and a higher bacterial load. Hence, these patients are at a higher risk of developing extrapulmonary TB (EPTB). It is also known that HIV/TB diagnosis is difficult due to the paucity of bacteria [6]. A negative smear for acid-fast bacillus, a lack of granulomas on histopathology, and a failure to culture mycobacterium tuberculosis (MTB) do not exclude an EPTB diagnosis [7]. Therefore, AIDS/TB blood-based diagnostic and/or monitoring methods need to be improved. 

Recently, several studies have investigated the critical role of matricellular proteins in infectious diseases. We have studied the roles of galectin-9 (Gal-9) and osteopontin (OPN) in dengue, HIV, and MTB infection [8,9,10,11]. Gal-9 induces apoptosis of T helper type (Th) 1 and Th17 cells through one of its receptor T cell immunoglobulin domain and mucin domain-3 (TIM-3). Conversely, Gal-9-TIM-3 interaction activates the bactericidal activity of MTB-infected macrophages [12]. Gal-9 levels in acute HIV infection were very high and declined rapidly by antiretrovirus therapy [10]. Recently, we have reported that plasma Gal-9 levels could identify viremic individuals with sensitivity and specificity of more than 90%, thus, they showed a potential to serve as a surrogate marker of viremia in HIV infected patients [13]. Furthermore, significantly elevated Gal-9 levels in HIV/TB coinfection, compared to HIV infection alone was reported [14]. It remains unknown if the same is observed when untreated AIDS and AIDS/TB patients are examined, because most medications can modify protein expression [15]. OPN is degraded by thrombin, matrix metalloproteinase (MMP)-3, MMP-7, and MMP-9, while Gal-9 is degraded by neutrophil elastase and MMP-3 [16]. These proteases are highly expressed at the inflamed lesion, as well as at the site of tissue remodeling. The measurement of these degraded matricellular proteins, together with their full-length (FL) form, may provide more in-depth pathophysiological information on the patients. 

To understand the evaluation value of matricellular proteins in inflammation and/or severity of HIV and/or MTB infection, we measured the FL and the cleaved form of Gal-9 and OPN in AIDS, TB, and AIDS/TB patients for the first time. For this purpose, we developed a new Gal-9 ELISA enabling detection of truncated Gal-9 (Tr-Gal9 ELISA). The patients in this study had no history of treatment medication and we were able to study samples at onset. In total, 38 cytokines/chemokines were also measured to understand their significance. 

## 2. Materials and Methods 

### 2.1. Study Design and Participants

This is a cross-sectional analytical study and the patients’ samples were collected at San Lazaro Hospital, in Manila, Philippines, from February 2019 to December 2019. Patients who were 19 years old or older were included in the study. Pulmonary TB (PTB), EPTB, or multi-drug resistant (MDR)-TB patients, and AIDS/TB patients were diagnosed based on clinical or bacteriological confirmation [17,18]. For bacteriological analysis, sputum microscopy and/or Genexpert were done and rifampicin resistant TB was regarded as MDR-TB, as previously described [19]. HIV infection was confirmed using Western blot in AIDS and AIDS/TB patients [18]. All AIDS patients had never been on antiretroviral therapy and were suffered from various opportunistic infections (OIs), such as pneumocystis pneumonia, tuberculosis, oral candidiasis, and central nervous system infections, such as cryptococcal meningitis and toxoplasmosis. Patients under antiretroviral therapy and anti-TB therapy, with hepatitis C or B virus infection, with known neoplasm or pulmonary diseases, or comorbid diseases were excluded. Each subject donated a 5-mL EDTA-treated peripheral-blood sample. EDTA-blood samples were centrifuged within 30 min of collection, and the plasma was stored at −80 °C until further analyses. A total of 30 normal human plasma samples, that are negative for HIV, hepatitis B and C viruses, were obtained from Bioivt (Hicksville, NY, USA).

### 2.2. Determination of FL-OPN and Ud-OPN

To identify FL-OPN, an ELISA kit (JP27158, IBL, Gunma, Japan) was used. Ud-OPN was determined using the Human OPN DuoSet ELISA Development System kit (DY1433, R&D Systems, Minneapolis, MN, USA) [9].

### 2.3. Determination of FL-Gal9 and Tr-Gal9

FL-Gal9 was measured using a human Gal-9 ELISA kit (GalPharma Co., Ltd., Takamatsu, Japan). The ELISA for Tr-Gal9 was constructed using two monoclonal antibodies against the N-terminal carbohydrate-recognition domain of human Gal-9, in which 9S2-3 (GalPharma) and biotinylated ECA8 (MBL, Nagoya, Japan) were used as the capture and the detection antibodies, respectively, together with streptavidin-conjugated horseradish peroxidase (Thermo Fisher Scientific, Waltham, MA, USA) for colorimetric detection. This ELISA demonstrated very similar detection profiles to those of Gal-9 ELISA from R&D Systems when the same sets of specimens were measured and compared (Appendix A). R & D Systems’ ELISA aberrantly overreacts against degraded Gal-9, compared to the intact one; hence, it practically only detects truncated Gal-9 with no quantification in the human specimens [20]. Tr-Gal9 ELISA also can detect degraded Gal-9 but without overreaction, therefore it is more quantitative. Gal-9 binds carbohydrates containing β-galactosides that is abundant in plasma as glycoproteins, which might interfere accurate measurement. We used 10 mM lactose in dilution buffers for specimen and detection antibody to prevent Gal-9 from forming complex with carbohydrates.

### 2.4. Determination of Cytokines/Chemokines

A total of 38 cytokines/chemokines were measured using Milliplex MAP assay kit (Millipore, Burlington, MA, USA, cat#HCYTMAG-60K), according to the manufacturer’s protocol. The data were acquired using a Bio-Plex 200 (Bio-Rad, Hercules, CA, USA) instrument and analyzed using Bio-Plex manager software (Bio-Rad) [21].

### 2.5. Ethical Considerations

This study adheres to the ethical considerations and ethical principles set out in relevant guidelines, including the Declaration of Helsinki, WHO guidelines, International Conference on Harmonization-Good Clinical Practice, Data Privacy Act of 2012, and National Ethics Guidelines for Health Research 2017. Prior to recruitment, written approval was secured from the Research and Ethics Review Unit of San Lazaro Hospital (SLH Peru 2018-012-1). Written informed consent was obtained from all patients prior to enrollment.

### 2.6. Statistical Analysis

Statistical analysis was performed using R Statistical Software (version 3.5.3; R Foundation for Statistical Computing, Vienna, Austria) and Prism 8 (Graphpad software, version 8.4.3). Mann-Whitney U test and Kruskal-Wallis test were used to assess the differences between the 2 groups and among multiple groups, respectively. The correlation between a set of data was examined using Spearman’s rank correlation coefficient. Receiver operating characteristic (ROC) curve analysis, including the corresponding area under the curve (AUC) calculation, was conducted to analyze the ability of biomarkers to discriminate between a selected pair of groups consist of AIDS, TB, AIDS/TB, normal subjects, deceased patients, and survivors.

## 3. Results

### 3.1. Clinical Findings

A total of 111 patients were enrolled, 5 patients were excluded, and 24 AIDS, 49 TB, and 33 AIDS/TB patients were studied (Figure 1). The study group had never been treated. The mean age of subjects was 33, 40, and 29 among HIV, TB, and HIV/TB patients, respectively (*p* = 0·001) (Table 1). Neither CD4 counts nor HIV viral load was significantly different between AIDS and AIDS/TB group, and there was no correlation between CD4 counts and HIV viral load in each group (Table 1, Figure 2). AIDS patients suffered from various OIs. The TB group had 43 (87.8%) patients who were bacteriologically confirmed and 6 (12.2%) were clinically diagnosed. Meanwhile, 17 (51.5%) patients were bacteriologically confirmed and 16 (48.5%) were clinically diagnosed in the AIDS/TB group. Additionally, 12 patients (28%) among the bacteriologically confirmed (TB group) had MDR-TB, while there was only 1 case of MDR-TB in the AIDS/TB group. Additionally, in both groups, most patients had PTB (43 TB (87.8%) and 24 AIDS/TB (72.7%)), while the rest had EPTB. The most frequent EPTB site are the lymph nodes, followed by gastrointestinal, CNS, pericardial, skeletal, and laryngeal TB.

### 3.2. Causes of Death

In total, 3 AIDS, 6 TB, and 5 AIDS/TB patients died within 12 weeks after enrollment. A total of 2 AIDS patients died of cryptococcosis. A total of 2 EPTB and MDR-TB patients died. In AIDS/TB group, 3 patients died of EPTB (Table 2).

### 3.3. ROC Analysis of Measured Molecules

To understand the discriminating power of matricellular proteins and cytokines/chemokines between the normal group and each disease group, ROC analysis was carried out and AUC values were expressed by heat maps (Figure 3A). In comparison with normal group, bright colors in the heat map which represents high AUC values are seen in matricellular proteins, interferon gamma induced protein-10 (IP-10) and tumor necrosis factor-α (TNFα), across three disease groups. A total of 13 molecules which demonstrated AUC values higher than 0.8 were listed as the candidates of disease markers (Table 3). The same ROC analysis was applied to examine if there are molecules that could discriminate a specific disease group among the other disease groups. FL-Gal9 and Tr-Gal9 discriminated TB from AIDS as well as TB from AIDS/TB by the AUC values higher than 0.8. Especially Tr-Gal-9 was noticeable with AUC values of 0.9991 and 1.0000 in TB vs AIDS and in TB vs AIDS/TB, respectively. On the other hand, we could not find any molecules with a remarkable difference between AIDS and AIDS/TB.

Each disease group includes several patients who died within 12 weeks after the enrollment. If some molecules distinguish them from survivors, they might serve as severity markers. ROC analysis identified several molecules, which were listed as the candidates of severity markers (Table 3). Clinical hematology/biochemistry data of patients were also fed for this severity marker identification. These candidates were expanded into beeswarm plots to show the distribution of each individual, and further examined if there are statistically significant differences compared to controls (Figure 4 and Figure 5).

### 3.4. The Levels of Gal-9 in Three Different Groups

The plasma FL-Gal9 of AIDS (median: 650 pg/mL), TB (median: 358 pg/mL), and AIDS/TB patients (median: 567 pg/mL) were all significantly (*p* < 0.0001) higher compared to normal controls (median: 55·5 pg/mL) (Figure 4A). Likewise, plasma Tr-Gal9 was significantly (*p* < 0.0001) higher in AIDS (median: 3820 pg/mL), TB (median: 1612 pg/mL), and AIDS/TB (median: 3654 pg/mL) patients, compared with the control (median: 416 pg/mL) (Figure 4A).

### 3.5. The Levels of OPN in the Three Different Groups

The plasma FL-OPN of patients with AIDS (median: 661 ng/mL), TB (median: 666 ng/mL), and AIDS/TB (median: 810 ng/mL) was significantly (*p* < 0.0001) higher compared to the control (median: 129 ng/mL) (Figure 4A). Likewise, plasma Ud-OPN was significantly (*p* < 0.0001) higher in AIDS (median: 123 ng/mL), TB (median: 65·1 ng/mL), and AIDS/TB (median: 103 ng/mL) patients, compared with the control (median: 8·34 ng/mL) (Figure 4A).

### 3.6. Correlations between Each Matricellular Protein

The correlation of the levels of matricellular proteins were studied. In all three groups, there were high or moderate correlations between FL-OPN and Ud-OPN (AIDS: *r* = 0.89, TB: *r* = 0.56, AIDS/TB: *r* = 0.69) and between FL-Gal9 and Tr-Gal9 (AIDS: *r* = 0.70, TB: *r* = 0.56, AIDS/TB: *r* = 0.41). In AIDS group, there was a moderate correlation between Tr-Gal9 and Ud-OPN (*r* = 0.42) (Appendix A).

### 3.7. Cytokines/Chemokines in the Three Different Groups

All the cytokines/chemokines selected by AUC values higher than 0.8 showed statistically significant differences against the controls in accordance with the results of ROC analysis (Figure 4A, Table 3). The followings are the confirmed candidates of cytokines/chemokines that might be useful as the disease markers; IL-10, TNFα, IL-8, IP-10, and macrophage derived chemokine (MDC) for AIDS; IL-2, TNFα, IL-8, and IP-10 for TB; and IL-1α, IL-10, IL-15, TNFα, IL-8, IP-10, and monocyte chemotactic protein-1 (MCP-1) for AIDS/TB.

### 3.8. Specificity and Sensitivity

The specificity and sensitivity of high AUC molecules (>0.8) were analyzed. In AIDS, Ud-OPN showed the highest specificity and sensitivity, and FL-OPN, Tr-Gal9, and IP-10 showed a specificity and sensitivity higher than 0.95 in all three groups. In TB, FL-OPN showed the highest value, and Ud-OPN and IP-10 values were higher than 0.95. In AIDS/TB, Tr-Gal9 value was the highest and FL-Gal9, FL-OPN, Ud-OPN, and IP-10 values were higher than 0.96 (Appendix A).

### 3.9. Subtype of TB

Both FL- and Tr-Gal9, and FL-OPN did not show any significant differences in MDR-TB and EPTB (data not shown). Ud-OPN was significantly (*p* = 0.0227) lower in MDR-TB, but higher (*p* = 0.0225) in EPTB (Figure 4B). This analysis was carried out not only against the candidates of biomarkers obtained by ROC analysis (Table 3) but all the factors measured in the current study, however only Ud-OPN demonstrated significant differences.

### 3.10. Markers Associated with Mortality

In AIDS group, 7 out of 11 candidates showed statistically significant differences between deceased patients and survivors. They are FL-Gal9, Tr-Gal9, IL-1 receptor antagonist (IL-1RA), IL-17A, transforming growth factor-α (ΤGFα), platelet counts, and creatinine (Figure 5A). In TB and AIDS/TB, all the nominated candidates were confirmed by the statistically significant differences (Figure 5B,C). They are IL-6, granulocyte-colony stimulating factor (G-CSF), MCP-1, neutrophil proportion, and monocyte proportion in TB, and macrophage inflammatory protein (MIP)-1β in AIDS/TB.

The accuracy of each molecule to predict mortality was calculated based on the ROC analysis (Appendix A). In AIDS, Tr-Gal9, IL-17A, and TGFα showed the high Youden index of more than 1.900. In TB, IL-6 was the highest by the Youden index of 1.698, and the only AIDS/TB candidate MIP-1β showed a Youden index of 1.764.

### 3.11. Effects of Age and Sex

The study groups were biased in age and sex (Table 1). Some factors demonstrated weak correlations with age, including IL-1RA; however, matricellular proteins did not (Appendix A). Creatinine was lower in females, in accordance with its known sex differences. Ud-OPN was also lower in females. Ud-OPN in TB was lower compared with AIDS or AIDS/TB, and the percentage of TB in the females (78%) was much higher than in the males (40%) in the current cohort, which explained the reason of lower Ud-OPN in females (Appendix A).

## 4. Discussion

Here, we asked whether FL or a mixture of FL and the cleaved form of Gal-9 or OPN could reflect the inflammation and/or the severity of AIDS or/and TB. The study group had never been treated. The results showed that the plasma levels of all matricellular proteins and some of cytokines/chemokines among each group were markedly elevated, when compared with normal samples. The heat-map analysis of AUC showed the brightest areas in matricellular proteins. It was evident that Tr-Gal9 could be the best marker to differentiate TB from AIDS and/or AIDS/TB among the 42 molecules examined.

The correlation analysis was conducted between matricellular proteins. The lack of correlation between FL-OPN and FL-Gal9 in all three groups indicates that they are synthesized using different pathological mechanisms. The moderate correlation between Ud-OPN and Tr-Gal9 in AIDS indicates the shared underlying mechanisms of the formation of the cleaved products. It is known that only Ud-OPN, but not FL-OPN, is negatively correlated with memory T cell numbers in MTB infection, indicating that Ud-OPN is responsible for the migration of memory T cells to granulomatous lesions [21]. These findings suggest that the degradation of OPN occurs via immune activation and the acquirement of a novel immune-related function of matricellular proteins. FL-OPN is known to be cleaved by MMP9 [22], MMP3 or MMP7 [23]. Neither FL- nor Ud-OPN were associated with the severity of the disease. The lower levels of Ud-OPN in MDR-TB, compared to non-MDR-TB, may be related with their protective function [24], as we have previously proposed that they may have a chemotactic activity on memory T cells, toward granuloma [21].

Gal-9 consists of two carbohydrate-recognition domains at the N- and C-terminus tethered by a linker peptide. Degradation at the linker peptide inactivates the protein, followed by their complete degradation. We found that one of the Gal-9 ELISA kits in the market aberrantly overreacts to these degradation products and falsely enhances the signal [20]. Hence the kit (from R&D Systems) has no use for the quantification of Gal-9; however, it might be useful to sensitively detect degraded Gal-9, whose concentration in the blood may reflect the pathophysiological conditions of the patients. Using well characterized monoclonal antibodies, we constructed a Gal-9 ELISA that could detect degraded Gal-9 without overreaction to the degraded products (Tr-Gal-9 ELISA), which was used to explore its potential utility. In the current study, Tr-Gal-9 measurement showed the best diagnostic power to distinguish between normal subject and AIDS, as well as to distinguish AIDS alive patients from deceased patients. We have reported that plasma Gal-9 levels could identify viremic individuals with sensitivity and specificity of more than 90%, and positively correlated with viral load and negatively correlated with CD4 counts in HIV patients on anti-retroviral therapy using the kit from R&D system [13]. We should clarify if our newly developed Tr-Gal-9 ELISA could be used as a better surrogate marker for the treatments of AIDS and AIDS/TB.

Other cytokines/chemokines, such as TNFα, IP-10, IL-8, and IL-10, which are also associated with a variety of inflammatory syndromes, were also found to be elevated. Among them, IP-10 showed the highest Youden index in all three groups. IP-10 plasma concentration was known to be the most significant contributor in a multivariate model of HIV infection. It was found to be synthesized by monocytes or dendritic cells through TLR7/8 [25]. In TB infection, it was claimed that IP-10 in combination with other makers, such as c-reactive protein (CRP), could optimize the triage performance of symptomatic patients [26]. In agreement with their study, it was shown that the Youden index of IP-10 was as high as those of matricellular proteins. But IP-10 was not found to be a severity marker here, though both FL- and Tr-Gal9 were found to be severity markers of AIDS. Therefore, Tr-Gal9 could be a better marker for the triage of patients. In the current study, we measured both FL- and Tr-Gal9 simultaneously for the first time. The levels of FL-Gal9 were measured previously and a median of 325 pg/mL was obtained in chronic HIV infection [27], which was lower than that obtained for AIDS in this study. Additionally, it was claimed that changes of Gal-9 levels were greater than those of CRP and serum amyloid A in acute HIV infection [10]. We found that Tr-Gal9 was elevated in all three groups and that their Youden index was higher than that of FL-Gal9. Therefore, the monitoring of Tr-Gal9 in HIV infected individuals would help to detect the occurrence of MTB and other OIs more sensitively at an early stage. This might eliminate the need for CD4 cell counts and viral load measurements, which would lead to a cost reduction. The elevations of Tr-Gal9 in AIDS and AIDS/TB may reflect a systemic inflammation associated with multiple organ failure, because we also found that platelet counts were significantly low (*p* = 0.0260) in deceased AIDS patients (Figure 5A).

It should be noted that FL-Gal9 levels are also extremely high in dengue patients (dengue hemorrhagic fever: 2464 pg/mL; dengue fever patients 1407 pg/mL, as compared with other febrile illness: 616 pg/mL; healthy: 196 pg/mL). The Gal-9 levels were inversely correlated with thrombocytopenia which is often associated with cytokine storm syndromes and could serve as a marker of disease severity [8]. The increase of Gal-9 were also seen in malaria and the levels were higher in severe malaria than in uncomplicated cases [28]. Likewise, Gal-9 elevations were observed in leptospirosis [29], acute HIV [10], influenza, and hepatitis virus infection [30]. Further investigation would be necessary whether Tr-Gal-9 could reflect hyperinflammatory/cytokine storm syndromes in these diseases. Very recently, the elevation of Gal-9 in COVID-19 pneumonia patients was reported using a Luminex assay system from R&D Systems. Like their Gal-9 ELISA, the Luminex also quantifies irrelevantly high concentration of Gal-9 [31]. It would be worth examining COVID-19 specimens using Tr-Gal9 ELISA, which responds degraded Gal-9 with similar detection profiles to the R&D Systems’ ELISA but more quantitative (Appendix A).

As demonstrated, plasma Gal-9 concentration could be an excellent surrogate marker of HIV infection [13], though the role of Gal-9 in the pathogenesis of HIV remains an open question by the conflicting reports where different in vitro assay systems were employed. Gal-9 at 100 nM stimulated infectivity of HIV in Jurkat T cells through cell surface protein disulfide-isomerase [32]. On the other hand, Gal-9 at 15 nM activated p21 expression and prevented HIV infection and propagation in phytohemagglutinin-stimulated CD4 T cells from healthy individuals [33]. Furthermore, Gal-9 at 500 nM reactivated latently infected HIV together with APOBEC3 induction in latently infected CD4 T cells from patients, which may lead to the eradication of HIV [34]. Concentration of Gal-9 to evoke these responses is far higher than the highest concentration of FL-Gal-9 observed in the current experiment: 2049 pg/mL in AIDS (59 pM). Therefore, the discussion of the role of Gal-9 must be based on the assumption that the local concentration of Gal-9 at the site of infection should be very high. We assume that a hike of plasma Gal-9 in the acute phase of HIV infection may be caused by a massive T cell death through bystander pyroptosis of infected individuals, which causes leakage of Gal-9 from the cytoplasm. Survived T cells may be activated by DAMPs and Gal-9 to induce p21 and ABOBEC3. These intrinsic immune mechanisms must have been protective against many retroviruses. Gal-9 is known to be degraded swiftly by proteinases, probably in order to prevent off-target effects of this multi-functional protein. In the current study, the diagnostic power was higher in Tr-Gal9 than in the active form, FL-Gal9. It is possible that the measurement of some degradation intermediates may provide more stable diagnosis than the measurement of active but non-stable FL-Gal-9.

According to the recent study in South Africa, IL-1RA, IL-6, IL-8, MIP-1α, MIP-1β, and IP-10 could segregate deceased patients from survivors in AIDS/TB [35]. Although the patients of our study had never been treated and the number of deceased patients was far lower than their study, we also nailed down MIP-1β as a severity marker of AIDS/TB. Our ROC analysis nominated IL-1RA, IL-8, and MIP-1α as candidates of severity marker of AIDS but not of AIDS/TB, though the results have some similarities.

Finally, because the majority of AIDS/TB patients reside in resource-limited places, the development of point-of-care testing, which measures Tr-Gal9 and Ud-OPN at the same time, would be important to evaluate the severity of the disease in patients with AIDS, TB, and AIDS/TB.

## 5. Conclusions

FL-OPN, Ud-OPN, FL-Gal9, Tr-Gal9, and 38 cytokines/chemokines were measured in the plasma from patients with AIDS, TB, and AIDS/TB. The study confirmed that Tr-Gal9 is the most sensitive biomarker of inflammation and severity in AIDS and AIDS/TB and proposed that its levels would be useful to monitor the development of AIDS and TB in HIV-infected individuals.

## Figures and Tables

**Figure 1 biomolecules-10-01495-f001:**
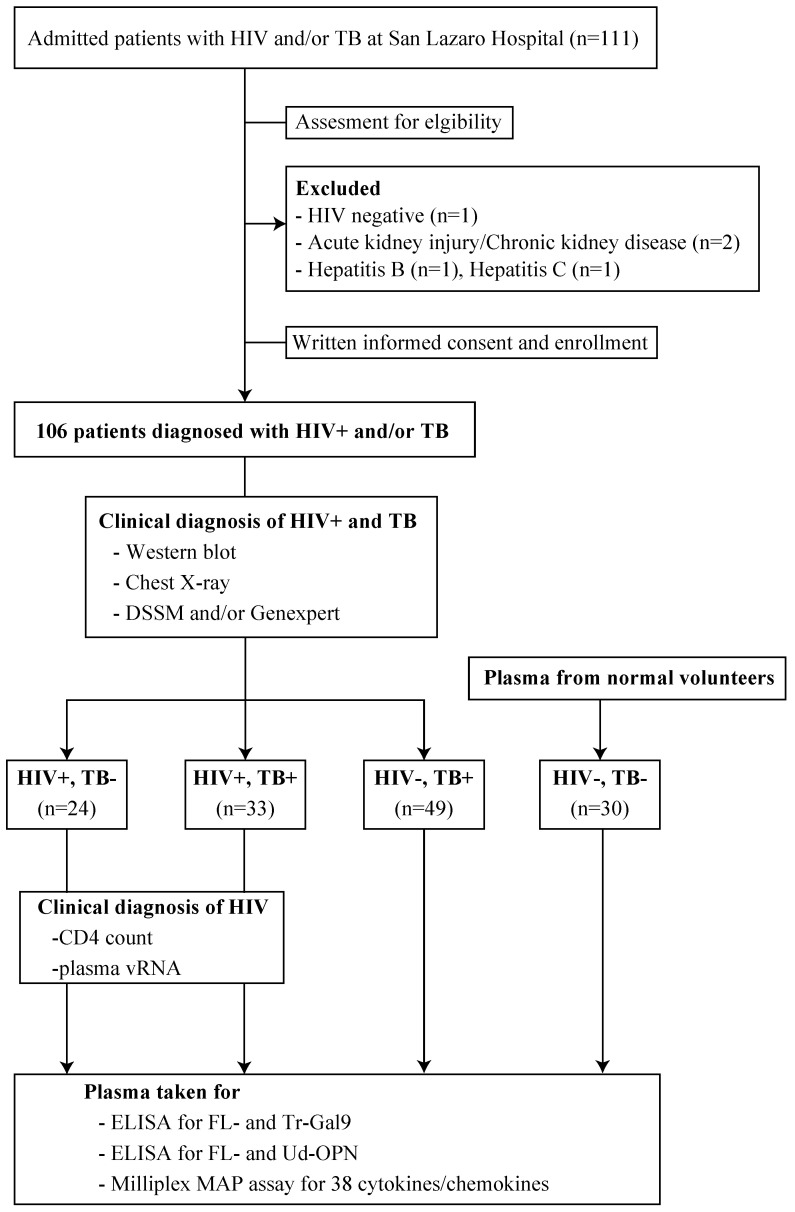
Study flowchart. Adult patients admitted with TB or HIV at San Lazaro Hospital were screened for eligibility. Eligible patients who met the inclusion criteria were randomly selected. A total of 106 patients were enrolled, 24 had HIV with a CD4 count of <200 cells/µL, 49 had TB, and 33 had AIDS/TB. Tuberculosis was microbiologically diagnosed using direct sputum smear microscopy and/or Genexpert, or clinically, using WHO signs and symptoms. All AIDS patients had never been on antiretroviral therapy and suffered from various OIs, as detailed in the Materials and Methods.

**Figure 2 biomolecules-10-01495-f002:**
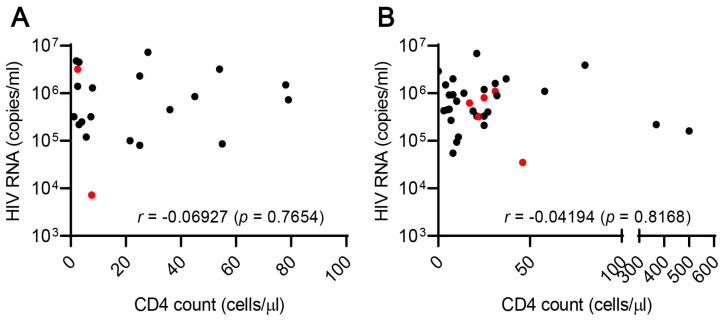
Correlations of CD4 count and viral load in patients with AIDS (**A**) and AIDS/TB (**B**). Red circles represent deceased patients.

**Figure 3 biomolecules-10-01495-f003:**
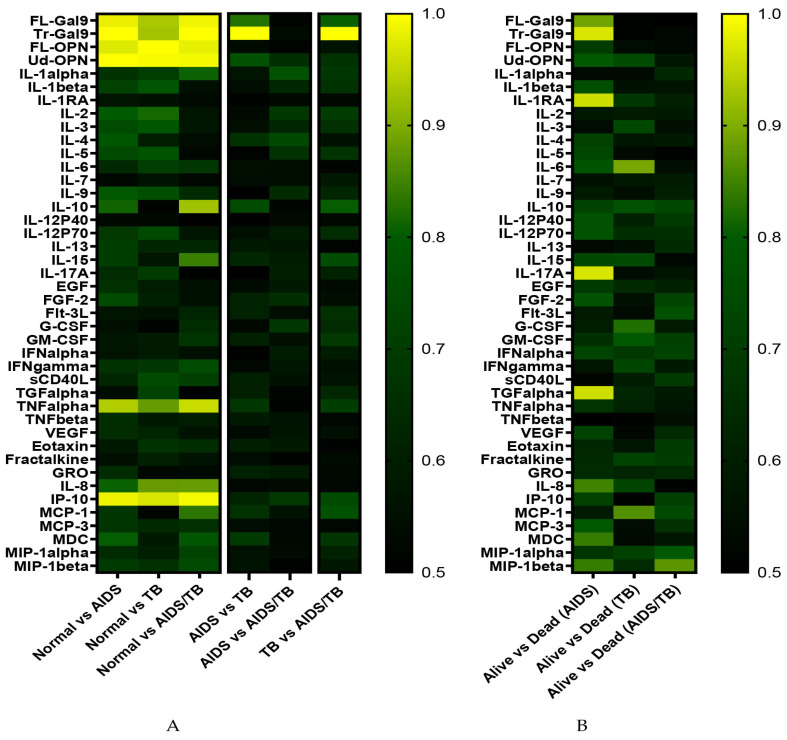
Fluctuations in matricellular proteins and cytokines/chemokines in the plasma of AIDS, TB, and AIDS/TB patients. Indicated groups were compared using ROC curve analysis. Heat maps were generated using the AUC values to identify plasma factors that may be useful for diagnosis. Comparison among normal subject, AIDS, TB, and AIDS/TB patients (**A**), or between living and deceased patients (**B**).

**Figure 4 biomolecules-10-01495-f004:**
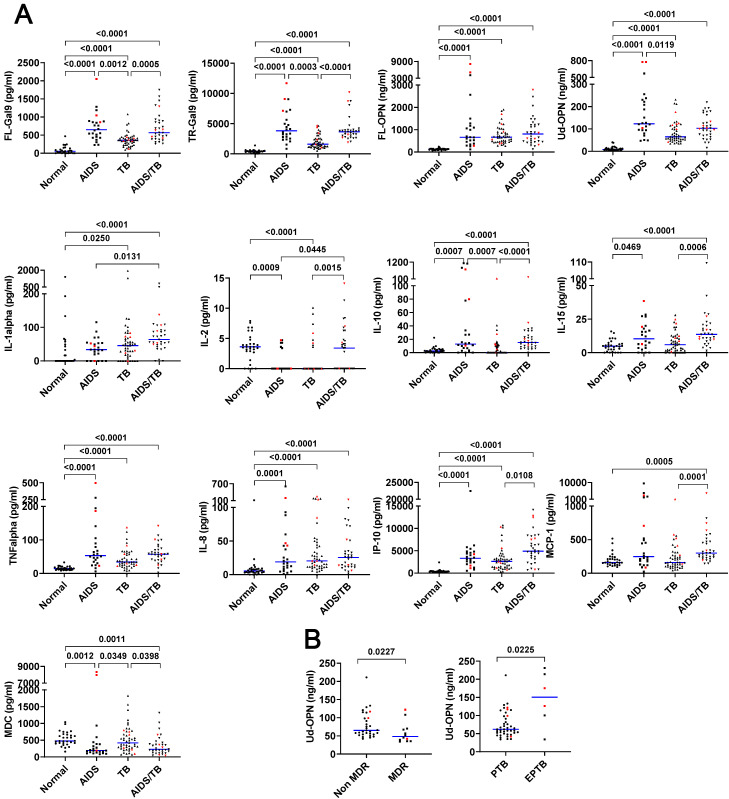
Amounts of plasma levels of selected molecules under different clinical conditions. (**A**) Candidates of disease markers selected by ROC analysis. (**B**) Ud-OPN differentiated subtypes of TB. Red circles represent deceased patients. When a statistically significant difference was found between the groups, the *p*-value was shown on the connecting line between the groups.

**Figure 5 biomolecules-10-01495-f005:**
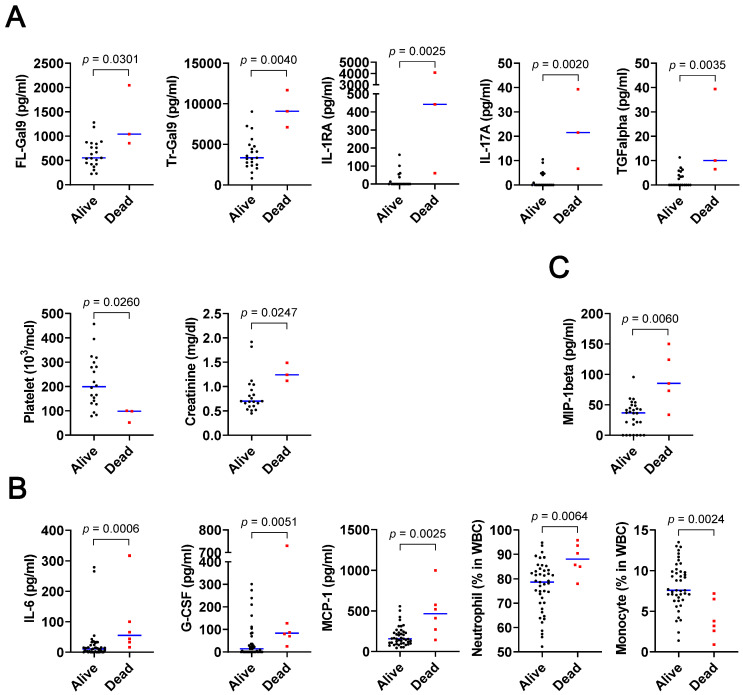
Molecules whose AUC values are > 0.8 and *p* < 0.05 in deceased patients, compared with survivors. AIDS (**A**), TB (**B**), and AIDS/TB patients (**C**).

**Table 1 biomolecules-10-01495-t001:** Characteristics of AIDS, TB, and AIDS/TB patients.

Group	AIDS N = 24	TB N = 49	AIDS /TB N = 33	*p* Value
**Age (Years), mean ± sd**	33.6 ± 8.1	40.1 ± 13.1	29.5 ± 6.5	0.001
**Gender, no.(%)**				
**Male**	22 (91.7)	35 (71.4)	31 (93.9)	0.0174
**Female**	2 (8.3)	14 (28.6)	2 (6.1)	
**CD4 count (μL)** **mean ± sd**	25.5 ± 21.3	-	47.4 ± 90.0	0.2787
**≤200 no.(%)**	21 ^#^ (100.0)	-	30 (93.8)	0.5123
**201-500 no.(%)**	0 (0.0)	-	2 (6.3)	
**>500 no.(%)**	0 (0.0)	-	0 (0.0)	
**Viral Load(/mL)** **mean ± sd**	1,438,216.7 ± 1,885,017.5	-	1,040,727.3 ± 1,353,485.6	0.3577
**Type of diseases, no. (%)**				
**PTB ***	-	43 (87.8)	24 (72.7)	0.0862
**EPTB ****	-	6 (12.2)	9 (27.3)	

* pulmonary TB, ** extra pulmonary TB; ^#^ data from only 21 cases were available.

**Table 2 biomolecules-10-01495-t002:** Clinical features of dead patients.

Patient No.	Age	Gender	Diagnosis	CD4 Count (/µL)	Virus Load(/mL)
**AIDS**					
**1**	27	M	Disseminated cryp *.	8	7.2 × 10^3^
**2**	32	M	BP **, PCP ^#^, Thrush	3	3.2 × 10^6^
**3**	24	M	CNS ^$^ cryp.	NA	3.0 × 10^5^
**TB**					
**1**	33	M	EPTB		
**2**	37	M	PTB, BP		
**3**	41	M	EPTB		
**4**	48	F	PTB (MDR-TB)		
**5**	52	F	PTB (MDR-TB), BP		
**6**	31	M	PTB		
**AIDS/TB**					
**1**	36	M	PTB, CNS Lymphoma, Thrush	46	3.5 × 10^4^
**2**	23	M	EPTB, Thrush, PPE ^&^	17	6.2 × 10^5^
**3**	25	M	EPTB, Thrush, PPE	NA	3.2 × 10^5^
**4**	35	M	PTB, Bacterial pneumonia, PCP, Thrush, PPE	NA	8.0 × 10^5^
**5**	20	M	EPTB, BP, PCP, Thrush, PPE	31	1.1 × 10^6^

*: Cryptococcosis, **: bacterial pneumonia, ^#^: pneumocystis pneumonia, ^$^: central nervous system, ^&^: popular pruritic eruption, EPTB and PTB are the same as Table 1.

**Table 3 biomolecules-10-01495-t003:** Candidates of disease specific and severity (death)-related biomarkers.

Type of Biomarker	AIDS	TB	AIDS/TB
Disease and severity ^#^	FL-Gal9, Tr-Gal9	----	----
Disease	FL-OPN, Ud-OPN, IL-10, TNFα, IL-8, IP-10, MDC	FL-Gal9, Tr-Gal9, FL-OPN, Ud-OPN, IL-2, TNFα, IL-8, IP-10	FL-Gal9, Tr-Gal9, FL-OPN, Ud-OPN, IL-1α, IL-10, IL-15, TNFα, IL-8, IP-10, MCP-1
Severity (death)	IL-8, MDC, IL-1RA, IL-17A, TGFα, MIP-1α, PC *, Creatinine, sGPT	IL-6, G-CSF, MCP-1, Neutrophils, Monocytes	MIP-1β

^#^ All candidates were identified by receiver operating characteristic (ROC) analysis. Disease markers and Severity (death) markers were chosen in comparison between normal and each disease group, and deceased patients and survivors in each group, respectively. FL-Gal9 and Tr-Gal9 were chosen in both as disease markers and severity markers in AIDS, hence they are listed as Disease and severity marker of AIDS. *: platelet counts.

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
