# Peer review of "Plasma Levels of a Cleaved Form of Galectin-9 Are the Most Sensitive Biomarkers of Acquired Immune Deficiency Syndrome and Tuberculosis Coinfection"

_biomolecules, 2020, doi:10.3390/biom10111495_

Round 1
Reviewer 1 Report
Plasma levels of a cleaved form of galectin-9 are the most sensitive biomarkers of acquired immune deficiency syndrome and tuberculosis co-infection
By Shirley T. Padilla and co-authors
In this study, which involves 106 patients with HIV (24) TB (49) and HIV/TB (33) the authors show that Tr-Gal-9 is a sensitive biomarker of inflammation and severity in AIDS and AIDS/TB and proposed that its serum levels would be useful to monitor the development of AIDS and TB in HIV-infected individuals.
Considering the impact that HIV and TB have at social level, as well as for the National Health Systems, the availability of soluble biomarker with potential diagnostic and prognostic value is very important.
However, although the paper is potentially interesting and well-written, it could be consistently improved.
Fluctuations in MCP and CC molecules in the plasma of AIDS, TB and AIDS/TB patients has been reported in detail in Figure 3, but not correlation has been provided with validate markers and other pathophysiological conditions like evolution of CD4/CD8 ratio in HIV and HIV/TB patients during disease progression.
Most important no evident clinical information on the disease progression is available, except for very few dead patients.
This means that this study was mostly designed “to fish” fluctuation of molecules during the disease without providing mechanistic information on the observed effects.
In other words, many soluble molecules can be detected at serum level (increased or decreased) in subjects affected by different diseases . In order to validate a potential diagnostic and/or prognostic biomarker a pathophysiological link with the disease should be provided.
Gal-9 is a lectin molecule able to modulate immunoresponse. For example Gal9 can activate CD4+ cells or induce apoptosis in different subpopulations of lymphocytes. It will be very interesting to know the status of CD4/CD8 ratio in HIV patients with increased serum levels of Galectin-9 and Tr-Gal-9 during disease progression. What is the biological effect of increased serum levels of Tr-Gal9 on immune cells in HIV and HIV/TB patients ?
This information may potentially contribute to clarify the apparent conflicting results reported in the literature (ref. 28-29-30) regarding the biological role of Gal-9 in HIV patients.
This point should be discussed in the manuscript.
- An other important issue to be discussed is: -
the authors report that “FL-Gal9 levels are also extremely high and are associated with thrombocytopenia in dengue patients, which is often associated with cytokine storm syndromes”. Is this a specific signature of dengue disease ?,
Is gal-9 increased in the serum of different infective diseases?
Is the gal9 increased serum level linked to some specific “signature” of the disease?
- FL-Gal9 was measured using a human Gal-9 ELISA kit. The ELISA for Tr-Gal9 was constructed using two monoclonal antibodies against the N-terminal carbohydrate-recognition domain of human Gal-9, in which mAb 9S2-3 and biotinylated mAb ECA8 were used as the capture and the detection antibodies, respectively, together with streptavidin-conjugated horseradish peroxidase for colorimetric detection.
Theoretically, this ELISA detects both FL and truncated Gal-9……..
The author should demonstrate, in selected serum samples, the presence of FL- and Tr- Gal9 in western blot analysis, to validate the ELISA results.

Author Response
For Reviewer 1
Thank you for your constructive and useful to improve our manuscript. Below is your questions and our answers. In this study, we used a two-step method of first screening candidates for disease and severity markers by AUC values calculated through ROC analysis, and then analyzing them using yet other statistical methods. To make this clearer, the Abstract, Sections 3.3, 3.7, 3.9 and 3.10 was significantly revised.
In this study, which involves 106 patients with HIV (24) TB (49) and HIV/TB (33) the authors show that Tr-Gal-9 is a sensitive biomarker of inflammation and severity in AIDS and AIDS/TB and proposed that its serum levels would be useful to monitor the development of AIDS and TB in HIV-infected individuals.
Considering the impact that HIV and TB have at social level, as well as for the National Health Systems, the availability of soluble biomarker with potential diagnostic and prognostic value is very important.
However, although the paper is potentially interesting and well-written, it could be consistently improved.
Q1 Fluctuations in MCP and CC molecules in the plasma of AIDS, TB and AIDS/TB patients has been reported in detail in Figure 3, but not correlation has been provided with validate markers and other pathophysiological conditions like evolution of CD4/CD8 ratio in HIV and HIV/TB patients during disease progress.  Most important no evident clinical information on the disease progression is available, except for very few dead patients. This means that this study was mostly designed “to fish” fluctuation of molecules during the disease without providing mechanistic information on the observed effects.
A1. Thank you for your comment. We did not try to monitor the markers during the clinical course and simply evaluated their clinical markers of untreated patients. It is known that the protein synthesis could be affected by antibiotics. We reported that quinolones significantly enhanced OPN secretion, https://www.ncbi.nlm.nih.gov/pmc/articles/PMC3370769/. We stressed that the study group had never been treated (lines 151,169-70) and most HIV patients (AIDS, AIDS/TB) had some opportunistic infections including tuberculosis. Accordingly, most patients have very low CD4 counts (Table 1) and there were no correlations between CD4 counts and viral load as shown in new figure (lines 154-5, Figure 2). We therefore introduced another study on Gal-9 in ART treated patients (Ref 13) and discussed on future study (lines 334-8).
Q2. In other words, many soluble molecules can be detected at serum level (increased or decreased) in subjects affected by different diseases. In order to validate a potential diagnostic and/or prognostic biomarker a pathophysiological link with the disease should be provided. Gal-9 is a lectin molecule able to modulate immunoresponse. For example Gal9 can activate CD4+ cells or induce apoptosis in different subpopulations of lymphocytes. It will be very interesting to know the status of CD4/CD8 ratio in HIV patients with increased serum levels of Galectin-9 and Tr-Gal-9 during disease progression. What is the biological effect of increased serum levels of Tr-Gal9 on immune cells in HIV and HIV/TB patients?
This information may potentially contribute to clarify the apparent conflicting results reported in the literature (ref. 28-29-30) regarding the biological role of Gal-9 in HIV patients. This point should be discussed in the manuscript.
A2. As mentioned in the Discussion, the pathophysiological link of increased plasma Gal-9 is still an open question. But it may be worthy to present the conflicting reports on the possible functions of Gal-9 in HIV infection. We described our opinion in the Discussion (lines 373-91).
- An other important issue to be discussed is: -
Q3. the authors report that “FL-Gal9 levels are also extremely high and are associated with thrombocytopenia in dengue patients, which is often associated with cytokine storm syndromes”. Is this a specific signature of dengue disease ?, Is gal-9 increased in the serum of different infective diseases? Is the gal9 increased serum level linked to some specific “signature” of the disease?
A3. Lines 360-8
The levels of Fl-Gal-9 in Dengue virus infection is extremely high and higher than non-dengue infectious diseases (360-8), and inversely correlated with platelet numbers. The levels are also increased in other infectious diseases such as malaria, leptospirosis, hepatitis virus and HIV. The levels were associated with the clinical severity in dengue and malaria.
Q4. FL-Gal9 was measured using a human Gal-9 ELISA kit. The ELISA for Tr-Gal9 was constructed using two monoclonal antibodies against the N-terminal carbohydrate-recognition domain of human Gal-9, in which mAb 9S2-3 and biotinylated mAb ECA8 were used as the capture and the detection antibodies, respectively, together with streptavidin-conjugated horseradish peroxidase for colorimetric detection.
The author should demonstrate, in selected serum samples, the presence of FL- and Tr- Gal9 in western blot analysis, to validate the ELISA results.
A4 We published western data using a set of plasma of acute liver failure (ALF) patients and healthy control (HC) that were also used to characterize Tr-Gal9 ELISA (supplementary Figure 1B). The attached (Figure B, C and a table) is excerpt from the publication. High protein concentration of plasma allows only 0.25 µL/well (~20 µg total protein) which is too small for detecting Gal-9 by western blot, antibody affinity column was used to concentrate Gal-9 from 150 µL plasma. Plasma A – H (Figure B) consists of HC (B, C, D, E), HC + recombinant Gal-9 (A) and ALF (F, G, H). Quantification of Gal-9 differed significantly by ELISA kit; either from R&D Systems (RDS) or GalPharma (GalP = same as FL-Gal9 ELISA). All ALF (F, G, H) were very high in RDS measurement but H was low in FL-Gal9 ELISA. In western blot, recombinant Gal-9s which differ from natural Gal-9 in the size, namely Gal-9(0) and Gal-9(N), were used to examine yield in column steps as well as to serve as standard for quantification (Figure C). Full-length Gal-9 (upper panel) was seen in A, F and G, also marginally in E, matching FL-Gal9 ELISA. The yield of Gal-9 through the purification step based on FL-Gal9 ELISA was 35%, 21%, 20% and 40% for A, E, F and G, respectively, which is acceptable considering the small volume and low concentration of Gal-9 (the details are in the publication). This result supports the integrity of FL-Gal9 ELISA. In RDS measurement, F, G and H contain >40,000 pg/ml of Gal-9 (>6,000 pg/150 µL), which deviates significantly with the small amount of Gal-9 detected by western blot. More than 96% of Gal-9 bound to the column according to the RDS measurement. In order to fill the gap, the elution yield from the column by SDS-sample buffer must be very low. The overall yield was 11%, 4%, 3%, 3% and 2% for A, E, F, G and H, respectively. We concluded from this western blot and other data that RDS ELISA is not quantitative (the details are in the publication).
When the Tr-Gal9 measurement of A – H is plotted against FL-Gal9, it resembles to RDS vs FL-Gal9 plot, in consistent with the strong correlation of Tr-Gal9 ELISA and RDS ELISA. The overall yield through the column step is 24%, 15%, 16%, 22% and 12% for A, E, F, G and H, respectively…, which is not so bad. At least the measurement is supposed to be closer to the genuine Gal-9 concentration in the plasma. Both Tr-Gal9 ELISA and RDS ELISA can respond to degraded Gal-9, which must be the reason of the strong correlation. We mentioned in the Materials and Methods that Tf-Gal9 ELISA is very similar to the RDS ELISA that aberrantly overreacts against degraded Gal-9. This makes wrong impression that Tr-Gal9 is also out of quantitative. We will revise this part accordingly (Line 118-120).
We think these western blot data which is mostly excerpts from our earlier publication is not necessary for this report especially when we add data clarifying the quality of Tr-Gal9 ELISA in Supplementary Materials.

Reviewer 2 Report
This paper studies biomarkers on blood of patients with HIV or TB or a combination of the two, and each is compared to the others and to normal using standard statistical methods and ROC-plots. One of the highlighted biomarkers is galectin-9 and its truncated (cleaved) form consisting of only one CRD. The study appears to be well done, and the results are clear. So only a few questions, suggestions:
Suggest to use better abbreviation than CC for cytokines and MCP for matricellular proteins.
CC is a subtype of cytokines and chemokines, and I don’t think all listed in this paper belongs only to this subtype.
MCP is similar to the name of the cytokines MCP-1 and MCP-3 in this paper, so it becomes confusing in the text and figures. For example, when text mentions MCP with AUC in Fig. 2, the only item in Fig. 2 containing MCP in the name are the cytokines, and not Gal-9 and OPN as the author really mean.
Line 30: should it be Tr-Gal-9 had the highest ability to differentiate TB from AIDS or AIDS/TB? The wording is unclear.
Table 1. There are 24 AIDS patients, but under CD4 counts onle 21 are shown and said to be 100%. Where are the other 3, or is it misprint.
Figures are out of sync with text in my downloaded pdf file. Fig. 2 overlaps with the text, so becomes somewhat unclear. Fig. 3 and 4 goes outside pages and has a big gap in between. Perhaps a local problem on my computer? But worthwhile checking.
Author Response
For Reviewer 2
Thank you for your constructive and useful to improve our manuscript. Below is your questions and our answers. In this study, we used a two-step method of first screening candidates for disease and severity markers by AUC values calculated through ROC analysis, and then analyzing them using yet other statistical methods. To make this clearer, the Abstract, Sections 3.3, 3.7, 3.9 and 3.10 was significantly revised.
This paper studies biomarkers on blood of patients with HIV or TB or a combination of the two, and each is compared to the others and to normal using standard statistical methods and ROC-plots. One of the highlighted biomarkers is galectin-9 and its truncated (cleaved) form consisting of only one CRD. The study appears to be well done, and the results are clear. So only a few questions, suggestions:
Q1 Suggest to use better abbreviation than CC for cytokines and MCP for matricellular proteins.
CC is a subtype of cytokines and chemokines, and I don’t think all listed in this paper belongs only to this subtype.
MCP is similar to the name of the cytokines MCP-1 and MCP-3 in this paper, so it becomes confusing in the text and figures. For example, when text mentions MCP with AUC in Fig. 2, the only item in Fig. 2 containing MCP in the name are the cytokines, and not Gal-9 and OPN as the author really mean.
A1. Thank you for the comment. We spelled out MCP to matricellular protein and CC to cytokines/chemokines.
Q2 Line 30: should it be Tr-Gal-9 had the highest ability to differentiate TB from AIDS or AIDS/TB? The wording is unclear.
A2. It was corrected according to your suggestions (line 32).
Q3. .Table 1. There are 24 AIDS patients, but under CD4 counts only 21 are shown and said to be 100%. Where are the other 3, or is it misprint.
A3. The CD4 counts from 3 AIDS patients were not available. It was described in the legend of Table1.
Q4. Figures are out of sync with text in my downloaded pdf file. Fig. 2 overlaps with the text, so becomes somewhat unclear. Fig. 3 and 4 goes outside pages and has a big gap in between. Perhaps a local problem on my computer? But worthwhile checking.
A4. We submitted PDF of the manuscript to avoid local problems.

Reviewer 3 Report
This is an interesting study involving the correlation of blood serum levels of tr-Gal9 and various CC chemokines in patients with AIDS and TB. Nevertheless, there are several major issues that must be resolved prior to consideration of publication in Biomolecules.
1) While the authors use an apparently non-specific antibody for trGal-9, it is apparently unknown what the epitope is. Gal-9 has two CRDs (one at the N and one at the C terminus), along with a linker peptide. The epitope is a crucial factor, e.g. what if it is the linker peptide that is cleaved from both CRDs? Does it bind to Gal-9N or Gal-9C? What is known about the epitope? This is all unclear and has not been sufficiently addressed in the manuscript.
2) Furthermore, related to this issue, is that CC chemokines have been reported to bind to galectins. And since the expression of these chemokines (e.g. IL8, IL10, IL1a, MIP, etc.) are modulated along with trGal-9, could it be that their interactions interfere with antibody binding and thus the readout in this study?
3) Tables 1,2,3 give various specs on the AIDS/TB patients, but not the normal patient group. This comparison needs to be made within each Table.
4) Also in the Tables SD and p values related to controls are absent. And when the word "significant" is used, a p value must be given. This too is absent in the manuscript. Overall, it is unclear whether there is sufficient power to make the conclusions the authors state.
5) Figures 2,3,4 are difficult to read, because they apparently slipped during creation of the pdf.
6) The N and C CRD in Gal-9 as related to trGal-9 are first mentioned in the Discussion on line 330. This should be mentioned in the Introduction, along with the antibody binding epitope.
Author Response
For Reviewer 3
Thank you for your constructive and useful to improve our manuscript. Below is your questions and our answers. In this study, we used a two-step method of first screening candidates for disease and severity markers by AUC values calculated through ROC analysis, and then analyzing them using yet other statistical methods. To make this clearer, the Abstract, Sections 3.3, 3.7, 3.9 and 3.10 was significantly revised.
This is an interesting study involving the correlation of blood serum levels of tr-Gal9 and various CC chemokines in patients with AIDS and TB. Nevertheless, there are several major issues that must be resolved prior to consideration of publication in Biomolecules.
Q1. While the authors use an apparently non-specific antibody for trGal-9, it is apparently unknown what the epitope is. Gal-9 has two CRDs (one at the N and one at the C terminus), along with a linker peptide. The epitope is a crucial factor, e.g. what if it is the linker peptide that is cleaved from both CRDs? Does it bind to Gal-9N or Gal-9C? What is known about the epitope? This is all unclear and has not been sufficiently addressed in the manuscript.
A1. Thank you for the comment. We should have disclosed more information to wipe off doubt in terms of the quality of Tr-Gal9 ELISA. We revised the Materials and Methods (lines 115-20), and added supplementary Figure 1 to show the specificity and characteristic of Tr-Gal9 ELISA.
Q2. Furthermore, related to this issue, is that CC chemokines have been reported to bind to galectins. And since the expression of these chemokines (e.g. IL8, IL10, IL1a, MIP, etc.) are modulated along with trGal-9, could it be that their interactions interfere with antibody binding and thus the readout in this study?
A2. Yes. We understand the risk by potential interaction of Gal-9 with other entities, which might interfere an accurate quantification. So, we use 10 mM lactose in the assay buffer to prevent the complex formation. For clarification, we revised the Materials and Methods accordingly.(lines 120-3).
Q3. Tables 1,2,3 give various specs on the AIDS/TB patients, but not the normal patient group. This comparison needs to be made within each Table.
A3; Normal patients were not enrolled in this study as shown in Figure 1. Thirty normal human plasma samples, that are negative for HIV, hepatitis B and C viruses, were obtained from Bioivt (Hicksville, NY, USA).(lines 101-2).
Q4. Also in the Tables SD and p values related to controls are absent. And when the word "significant" is used, a p value must be given. This too is absent in the manuscript. Overall, it is unclear whether there is sufficient power to make the conclusions the authors state.
A: p values were given in red.
Q5. Figures 2,3,4 are difficult to read, because they apparently slipped during creation of the pdf.
Sorry for this mistake. We made a PDF of version of text.
Q6. The N and C CRD in Gal-9 as related to trGal-9 are first mentioned in the Discussion on line 330. This should be mentioned in the Introduction, along with the antibody binding epitope.
A6.
To all
In this study, we used a two-step method of first screening candidates for disease and severity markers by AUC values calculated through ROC analysis, and then analyzing them using yet other statistical methods. To make this clearer, the Abstract, Sections 3.3, 3.7, 3.9 and 3.10 was significantly revised.

Round 2
Reviewer 3 Report
The authors have adequately addressed previous concerns and critiques.